# Microstructure, Texture and Mechanical Properties in Aluminum Produced by Friction-Assisted Lateral Extrusion

**DOI:** 10.3390/ma14092465

**Published:** 2021-05-10

**Authors:** Viet Q. Vu, Laszlo S. Toth, Yan Beygelzimer, Yajun Zhao

**Affiliations:** 1Laboratory of Excellence Design of Alloy Metals for Low-Mass Structures, CNRS, University of Lorraine, 57070 Metz, France; vuquocviet84@gmail.com; 2LEM3 Laboratory, CNRS, Université de Lorraine, 57070 Metz, France; 3Thai Nguyen University of Technology, 666, 3/2 Street, Thai Nguyen 250000, Vietnam; 4Donetsk Institute for Physics and Engineering named after O. O. Galkin, National Academy of Sciences of Ukraine, pr. Nauki 46, 03028 Kyiv, Ukraine; yanbeygel@gmail.com; 5School of Materials Science and Engineering, Dalian Jiaotong University, 794 Huanghe Road, Dalian 116028, China; zhaoyajun@live.fr

**Keywords:** friction-assisted lateral extrusion process (FALEP), severe plastic deformation (SPD), aluminum Al-1050, grain refinement, simple shear texture, Lankford parameter

## Abstract

The Friction-Assisted Lateral Extrusion Process (FALEP) is a severe plastic deformation (SPD) technique for producing metal sheets from bulk metal or powder in one single deformation step at room temperature. In the present work, aluminum Al-1050 was deformed by FALEP. Then, its microstructure was examined by EBSD; the crystallographic texture by X-ray; material strength, ductility, and the Lankford parameter by tensile testing; the latter also by polycrystal plasticity simulations. It is shown that the microstructure was highly refined, with the grain size reduced more than 160 times down to 600 nm under the imposed shear strain of 20. The obtained texture was a characteristic simple shear texture with a shear plane nearly parallel to the plane of the sheet. The yield and ultimate strengths increased by about 10 times and three times, respectively. The Lankford parameter was 1.28, which is very high for aluminum, and due to the specific shear texture, unusual in a sheet. All these exceptional characteristics of Al-1050 were obtained thanks to the efficiency of the FALEP SPD process, which is a promising candidate for industrial applications.

## 1. Introduction

Ultrafine-grained (UFG) materials have gained more attention as they offer high strength and other outstanding service properties such as high wear resistance, good fatigue strength, superplasticity, etc. [1]. A possible avenue for producing UFG materials has been realized by using severe plastic deformation (SPD) techniques, which are metal-forming methods that impose a very large plastic strain on the processed sample. Not only UFG structures are created in SPD; other remarkable advantages, such as the suppression of micro-cracks and voids, are achieved thanks to the large hydrostatic pressure existing in almost all SPD techniques.

Some typical examples of SPD techniques for producing UFG bulk materials are equal channel angular pressing (ECAP), non-equal channel angular pressing (NECAP), high pressure torsion (HPT), and twist extrusion [2]. These techniques are effectively employed to produce bulk UFG materials. Initially sheet materials can be processed by the following SPD techniques: accumulative roll bonding (ARB) [3], asymmetric rolling (AR) [4], con-shearing [5], repetitive corrugation and strengthening [6], and equal channel angular rolling [7].

It is also possible to produce sheet metal from a bulk piece. Recently, we proposed a new SPD process named plastic flow machining (PFM, patented [8]) to address this issue. PFM is an extrusion-based technique in which a surface layer of a bulk sample is separated under a large hydrostatic pressure by means of a controllable lateral extrusion process to transform the bulk into a sheet (or fin) with UFG microstructure. The sheet or the fin produced by this process exhibits superior balance between strength and ductility and greater formability compared with other SPD counterparts, thanks to the large strain and the simple shear texture created during processing [9,10]. Mechanical modeling and texture modeling of this process can be found in [11,12].

Nakamura et al. [13] proposed the Friction-Assisted Extrusion (FAE) process to produce sheet metal. In their first work, a powder composed of pure aluminum and SiC ceramics was compacted into a bulk shape, then deformed by FAE with very large extrusion ratios to form thin strips. In a second study [14], both bulk and powder aluminum materials were deformed by FAE with extremely large extrusion ratios to produce thin strip materials with significant mechanical strength. The strips made of powder exhibited higher strength than those from bulk material. In both studies of the FAE process [13,14], the authors show that the unique advantage of FAE is the considerably lower compression pressure needed in FAE compared to conventional extrusion processes at the same extrusion ratios, thanks to the assistance of the friction force.

Inspired by the success of our PFM process, we designed and successfully tested the FAE process. For better clarifying the exact nature of FEA, we propose to call this process the Friction-Assisted Lateral Extrusion Process (FALEP). Compared to PFM, FALEP transforms not only a surface layer but the entire bulk sample into a sheet. Moreover, microstructure produced by FALEP is homogeneous, which is different from the PFM process, where there is a large gradient in strain amplitude, texture, and also in microstructure. When compared to conventional extrusion processes, where friction is usually detrimental, FALEP uses friction as a driving force for the plastic flow, turning the negative effect of friction into a positive one. Yet another key advantage of FALEP is that extremely high plastic deformation can be achieved; so high that the grain refinement process reaches the saturation state. Therefore, the minimum grain size can be achieved in a single operation. To date, only HPT processing is capable of producing such microstructures with an extremely large imposed strain in a single operation, but the sample size is limited in HPT. Although some upscaling efforts have been made to increase the sample size to 30 mm in diameter, it is typically limited to 10 mm [15], which is a barrier for industrial use. ECAP and its variants, including ECAP-conform [16] for producing continuous sheet samples, are able to produce larger size samples, but the shear strain imposed on the sample in each pass is relatively small; it is typically around 2 [17], while a shear of at least 10 is needed to reach the steady state grain size regime. Therefore, many passes are required in ECAP, which is not efficient from an industrial point of view. Rolling-based SPD techniques, such as ARB and AR, although capable of producing large size samples in a continuous manner, share the same limitations as ECAP and its variants; they cannot introduce a very large imposed strain in one pass. In addition to that, due to the relatively low hydrostatic pressure in rolling, material fracture can readily occur during processing [18]. The FALEP process used in the present study has the unique advantage of not only producing large-size samples but also imposing an extremely large strain in one single pass while providing high hydrostatic pressure during processing. Therefore, FALEP can be a satisfactory solution to the above existing problems of current SPD processing. Another important advantage of this process, which is featured in the present study, is that it produces simple shear textures with shear plane almost parallel to the plane of the produced sheet, which is a unique texture in sheets.

This paper presents a comprehensive study on FALEP, which was implemented on an existing ECAP machine. We examined various aspects, such as strain estimation, microstructure, texture, mechanical properties, and the Lankford parameter of the sheet produced from a commercial pure aluminum (Al-1050) bulk sample. The purpose of the present paper is to reveal the capabilities of the FALEP process regarding significant microstructure refinement, producing a unique texture that produces high Lankford value for aluminum sheets, and significantly improving mechanical strength. The process also has potential for upscaling to industrial conditions.

## 2. Materials and Methods

### 2.1. The FALEP Process

FALEP is an extrusion process; it requires two punches that are oriented perpendicular (see Figure 1). They are called normal and driving punches. The billet is fitted into the square-section channel (*w* × *w*) of the normal punch. While the top part of the sample is in contact with the normal punch, its bottom is lying on the top surface of the driving punch, which is sand blasted for having a rough surface. During extrusion, the sample material flows out laterally, between the driving punch and the die-wall. The sheet is formed within this gap, with a thickness *t* and width *w*. During processing, the sample is pressed by a pressure *P_N_* by the normal punch and the driving punch is moving at a constant speed *V*_0_.

It is important to emphasize that by moving the driving punch, a friction force *F_fr_* is applied on the bottom of the sample in the direction of its extrusion, thus helping the material to flow into the gap. Therefore, the friction plays a positive role; it is a driving force for the extrusion. Note that in other SPD processes, such as ECAP and NECAP, the friction force plays a negative role because it is acting opposite to the material flow, so hindering the process. A photo of the die-part of the experimental FALEP setup is shown in Figure 2.

The plastic strain applied in the FALEP process can be estimated in different ways, depending on the adopted strain mechanism. If one considers the process as a variant of non-equal channel angular extrusion (NECAP), the following formula gives the shear strain [19,20]:(1)$$\gamma =\frac{p}{c}+\frac{c}{p}$$
where *p* is the diameter of the entering channel and *c* is the thickness of the exit channel. In our experimental setup, p=w=20  mm and c=t=1  mm, which leads to γ=20.05.

Another way of looking at the FALEP process is that the bottom layer is moved by simple shear. To obtain the shear value, one has to consider that the material from the left top point of the layer has to reach the exit point, so it has to move by *w* = 20 mm. As the layer is 1 mm thick (*t*), the shear strain is: γ=w/t=20. This strain is so high that the material reaches the so-called steady state, where the grain size is minimum and remains constant [21]. It is actually equivalent to the strain that one could obtain in 10 ECAP passes.

### 2.2. Experimental Conditions

Commercially pure Al-1050 was selected for this study because it is frequently used in applications, for example, in electricity, and it is stronger compared to pure Al due to the 0.5% impurity content. Further strengthening, however, is not possible by heat treatments, contrary to other Al alloy systems, so the FALEP SPD process was selected to achieve UFG grain size, which increases the yield stress by the Hall–Petch effect.

The Al-1050 material was in the form of a square-billet with a cross section of 20 mm × 20 mm. Before processing by FALEP, the samples were annealed at 380 °C for 1 h and cooled in the furnace to improve the homogeneity of their microstructures. They were then subjected to the FALEP by adapting a three-axis ECAP machine for this purpose (Figure 2a,b), which is controlled by a hydraulic system. The maximum force capacity of each hydraulic punch was 72 tons, and the testing was done at room temperature. The die and the punches, as well as the hard block used for this process, were made of a high-alloyed Z160CDV12 steel, which was heat treated to obtain a hardness of 58 HRC and a yield limit of approximately 2 GPa. The vertical channel was square shaped with dimensions of 20 mm × 20 mm, while that of the horizontal channel was set at a thickness of 1 mm in order to establish a substantially large extrusion ratio of 20. During processing, the ND punch was controlled to press at a constant pressure *P_ND_* = 450 MPa (18 tons). The dimension of the driving punch was 20 mm × 19 mm and it was controlled to move at a constant speed of *V*_0_ = 3 mm/s. The pressing pressure *P*_0_ acting on this punch increased progressively to the maximum value of 300 MPa (12 tons) by the time the experiment finished. The total displacement of the driving punch was 60 mm, which was equal to the length of the produced in Figure 2c. This indicated that there was no sliding between the sample and the punch, meaning that friction with full adherence was generated at the interface between the punch and the sample. An optical micrograph of the bottom surface of the sample is shown in Figure 2c. This surface was observed to be fairly smooth and homogeneous; its rugosity was only about 10 µm. This is because there was no sliding between the bottom surface of the sample and the top surface of the punch thanks to the sticking friction between these two surfaces, generated by the normal pressure. The experiments were repeated many times (at least 10 times) and the very same results were obtained.

The microstructure and the crystallographic texture were examined by Electron Back Scattering Diffraction (EBSD), using a JEOL JSM-6500F field-emission gun-scanning electron microscope (Tokyo, Japan). For that purpose, the sample was first mechanically polished up to 4000 SiC grit papers and then electrolytically polished with a solution consisting of 90% ethanol and 10% perchloric acid at 20 V for 15 s at 20 °C. The orientation distribution function (ODF) of the texture was computed from the inverse pole figure maps of the EBSD measurements.

In order to examine the role of the driving friction in FALEP, we carried out additional experiments. A test was conducted without moving the driving punch and without sand blasting it. Figure 3 shows the test configuration and a sample that was partially extruded. In this second test (could be called T-NECAP), the extrusion process occurred at both sides of the sample and shaped it in the form of a letter T upside down. The results obtained from this test showed that when the pressure on the ND punch reached 1 GPa (40 tons)—which was more than twice as high as for the FALEP process—the fin length was only about 6 mm (Figure 3b). This means that even with a more than twice higher pressure on the ND punch compared to FALEP, T-NECAP still failed to deform the sample to produce sufficient fin length. This test clearly proved that without the movement of the driving punch—which produces “driving” friction to support the deformation process—the T-NECAP technique for the case of a large extrusion ratio (20 in this setup) is not feasible. It requires extremely high pressure and the extrusion stops, producing only a short fin.

## 3. Experimental Results

### 3.1. Microstructure Evolution

The initial microstructure of the Al-1050 sample consisted of coarse grains with an average grain size of about 100 µm. After the FALEP process, the microstructure of the produced fin exhibited significant grain refinement, with a grain size reduced more than 160 times to 600 nm, with relatively equiaxed grains (Figure 4). In addition, a very homogeneous microstructure was produced. The grain sizes obtained in different regions across the fin thickness, including the top, middle, and bottom areas of the fin, were about the same: 600 nm. This spectacular achievement of grain refinement was possible by the large strain imposed in the microstructure during this process; indeed, the minimum limiting grain size was attained in Al (room temperature deformation).

The uniformity of the microstructure can also be verified in the next-neighbor grain disorientation distributions presented in Figure 5: they are practically identical across the thickness of the fin. They also approach the distribution corresponding to the random orientation distribution case. The difference between the random (Mackenzie) distribution and the experimental one is due to the applied large deformation, during which a non-random crystallographic orientation distribution developed (called texture). The fraction of large angle boundaries was very high, about 92%, which was obtained by integrating the disorientation distribution in Figure 5 for angles larger than 15°, then multiplied by 100 (the total integral is 1.0). This fraction is the same value as for a shear strain of about 20 in Al-1050 deformed by other shear processing [22].

### 3.2. Crystallographic Texture

The initial texture of the Al-1050 samples is shown in Figure 6a; it was nearly random. However, after FALEP processing, a simple shear texture was achieved in all parts of the fin including top, middle, and bottom areas across the fin thickness (see Figure 6c,d). The texture is presented in the (x = FD, y = ND, z = TD) sample reference system. The shear plane SP was almost parallel to the plane of the sheet (Figure 6c); from the texture, it was estimated to be at ~3° with respect to FD. This value agrees well with the orientation of the ideal shear plane corresponding to the NECAP process, which is arctg(t/w)=2.86°. The strongest component in the texture is the C ({110} <001>), which corresponds to equal double slip [23]. The A1 and A2 components are also relevant, presenting much less intensity. The A component is present with low intensity. Among the ideal components of shear textures, the B and Bb components are the weakest. Finally, one can notice the presence of the 45° rotated cube (R_C_) component, which is not an ideal orientation of shear textures, but it is actually the second-most relevant component of the texture after C.

### 3.3. Mechanical Properties

Tensile tests were carried out at room temperature at a strain rate of 0.001 s^−1^ on the initial sample and on the 1.0 mm thickness fin at 0°, 45°, and 90° directions with respect to the FD direction, which is the longitudinal direction of the fin. The test results are presented in Figure 7a,b together with the geometry of the tensile specimens (Figure 7c). As seen in Figure 7b, the produced fin exhibited significant variation in yield strength as a function of testing direction; between 118 and 135 MPa (see Table 1). These values are about 10 times higher than the yield stresses before SPD processing, which were about 10–13 MPa. The uniform elongations, however, dramatically decreased after FALEP, which is a common feature of the SPD techniques and cold working processes in general.

### 3.4. Lankford Parameter

Two factors determine the formability properties of metallic sheets during deep drawing: strain hardening capacity and crystallographic texture. The effect of the crystallographic texture on formability can be readily measured by the Lankford parameter (also called *R*-value), which is the ratio of the width to thickness strain in tensile testing. *R*-value measurements were carried out on the Al-1050 fins produced by the FALEP process. The digital image correlation technique was used to simultaneously measure width and thickness strains during the tensile tests. Individual Lankford values in different directions (0°, 45°, and 90°) with respect to FD were measured just before necking. The average Lankford value ($\bar{R}$) and the planar anisotropy coefficient ($\Delta \bar{R}$) were calculated from the three individual Lankford values, they are defined by:(2)$$\bar{R}=\frac{R_{0}+2R_{45}+R_{90}}{4}\;\Delta \bar{R}=\frac{R_{0}-2R_{45}+R_{90}}{2}$$

The results are shown in Table 2. As can be seen, the average $\bar{R}$-value was 1.28, which is exceptionally high for aluminum sheets. However, there is planar anisotropy, expressed by $\Delta \bar{R}$ = −0.31.

## 4. Discussion

This paper presents a detailed study on how one can change the mechanical properties and the microstructure of a commercially pure Al metal using the FALEP process. The objective of this paper is two-fold: to show an example of the effectiveness of the FALEP for processing a frequently used aluminum alloy (Al-1050), and to promote the FALEP process itself for future applications in the SPD scientific community and in industry. The present Discussion Section deals with these two objectives.

### 4.1. FALEP Characteristics

It is clear that FALEP is a one-step process and capable transforming the microstructure into an UFG structure. Another great advantage of FALEP is that the extrusion forces are reduced with respect to other SPD processes. This was demonstrated by comparing the forces between FALEP and T-NECAP. The reason for the relatively low pressures needed for FALEP is that the loading state is multiaxial: there is an axial stress σ applied by the normal punch and a shear stress τ by friction. Assuming that the material obeys the von Mises yield criterion, the resultant flow stress $\bar{\sigma }$ during plastic flow is:(3)$$\bar{\sigma }=\sqrt{\sigma^{2}+3\tau^{2}}$$

It is clear that by adding a shear stress by friction, the normal stress needs to be decreased to reach the same value of the material flow stress. The following estimation can be done for a quantitative estimation of the contribution by friction. As there is adherence between the sample surface and the driving punch, we can approximate the friction coefficient to be about μ=1. The friction force is coming from the applied normal load, therefore: τ=μσ≅σ. Substituting this into the von Mises formula above, we obtain for the FALEP test: $\sigma =\bar{\sigma }/2$. Indeed, this was the ratio found between the FALEP and NECAP experiments for the normal load needed for the plastic deformation of the sample.

Another positive point of FALEP is that the deformation is uniform within the produced sheet. The uniformity of deformation can be concluded from the extreme uniformity of the microstructure, from the uniform grain shapes, the uniformity of the texture, and the precise uniformity of the disorientation distribution. Namely, if all these experimental features are all uniform, it means that the deformation is also uniform, because all these features depend on the amount of strain. Comparing this property to other SPD processes, we find that there is no other SPD process where such uniformity can be achieved. For example, comparing to the two most popular SPD techniques, ECAP and HPT, one finds a gradient in the amount of strain in ECAP perpendicular to the sample axis, and in HPT, there is a gradient in the radial direction of the disk.

Perhaps the most important characteristic of FALEP is the extremely large strain in a single step. As there is no estimation for that in previous papers on FALEP, two evaluations were presented in the present work. One was based on the similarity of the process with NECAP, which provided an analytical formula (Equation (1)), and the other assumes simple shear of the bottom layer of the sample. Both methods gave the same result, which in the present FALEP geometry, was a shear strain of 20. Such a high strain brings the material into the steady state regime of grain fragmentation in which the average grain size does not change any more [27]. This is a very positive achievement of the FALEP process because the minimum grain size can be reached in a single SPD operation.

Finally, another characteristic of FALEP is that it is a viable process to obtain a sheet from a bulk material in a single step, by extrusion. Nevertheless, in this regard, FALEP cannot compete with the usual rolling process, as rolling is a simpler process to obtain a sheet from a bulk metal. One aspect, however, is in favor of FALEP, which is the amount of strain. As shown above, it was about 20 in the presented FALEP configuration, which would be 11.5 equivalent von Mises strain in rolling. It is virtually impossible to reach such a high strain in rolling; even 99% thickness reduction (which can only be reached in many passes) is less than that; it is 4.0. Thus, rolling cannot be qualified as an SPD process because it is not possible to reach the steady state of grain fragmentation by rolling. Nevertheless, ARB-type rolling is an SPD process, because unlimited strain can be obtained. However, as many as 16 ARB steps are needed to reach the same strain, which is obtained in FALEP in a single pass.

### 4.2. Mechanical Behaviour of Al-1050

The experimental results obtained after FALEP of Al-1050 demonstrate well the material processing capacity of FALEP. The material strength increased by several factors: the yield stress 10 times, the ultimate tensile stress three times. A comparison of the mechanical properties between FALEP and other SPD processes for the same material Al-1050 is shown in Table 3. As can be seen, comparing with some well-known high-performance extrusion- and rolling-based SPD processes, such as ECAP, PFM, Incremental ECAP, ARB, and ASR, FALEP shows equal or greater capability in improving mechanical strength. Moreover, in terms of deformation steps, FALEP is more efficient, since only one pass is needed to achieve these superior mechanical properties, unlike other processes which require multiple passes for obtaining a similar performance. Nevertheless, there is one process that leads to higher yield and ultimate strengths in Al-1050, the HPTT (High Pressure Tube Torsion, [28]). HPTT is a tube-SPD process, where the tube is completely constrained by two mandrels, so the hydrostatic stress is very effective. One possible interpretation of the much higher mechanical strength in HPTT can be the relatively low number of vacancies created during HPTT; their creation is hindered by the hydrostatic stress. Due to this, after HPTT, in a tensile testing, there would be small number of vacancies, which would be needed for the non-conservative motion of dislocations so very high yield stress can be achieved. In order to verify this hypothesis, vacancy density measurements should be done on the deformed material.

While material strength is increasing multiple times in SPD processes, ductility reduces to low values. This is the well-known strength-ductility paradox, which is in a focus of interest in material science. FALEP is not an exception, but nevertheless, compared to other processes (see Table 3), both the uniform and total elongations are significantly better than for other processes listed in Table 3, except the PFM process, where the values are similar.

Concerning grain refinement, the comparison between FALEP and other processes in Table 3 shows that FALEP is producing an average grain size similar to other SPD processes in Al-1050. This is the consequence of reaching the steady state, which leads to the same grain size for all processes that are able to bring the material state into the steady state. An important condition, however, is that the alloy composition and the processing temperature have to be same for all processes that are compared [21].

### 4.3. Crystallographic Texture

As displayed in Figure 6, the measured deformation texture is a typical shear texture, characteristic to aluminum, which develops a strong C component. While all other ideal components also appeared in the texture, their intensity was low compared to C. As indicated in Section 3.2 above, it is the 45° rotated cube (R_C_) component that was the second-strongest texture component. However, the cube is not a stable component in shear; it is continuously rotating with the rigid body spin inherent in the simple shear velocity gradient tensor [33]; thus, the cube is totally unstable in simple shear. Its apparent stability in the 45° rotated position is actually the sign of dynamic recrystallization (DRX). The same observation was made in simple shear of copper, where the 45° rotated cube appeared and was simulated using a DRX model [34]. Indeed, when plastic deformation is so high that the steady state of grain fragmentation is reached, the continuous DRX process is active, as proposed in several studies. As can be seen from the above reasoning, the crystallographic texture can show the signs of different metallurgical mechanisms, such as DRX, so it can be used as a mean for identification of the occurrence of DRX. From this, we can conclude that DRX took place during FALEP in Al-1050 at room temperature.

### 4.4. Lankford Parameter

It is widely known that aluminum and aluminum alloy sheets processed by conventional rolling and subsequent annealing present very low *R*-values (between 0.5 and 0.85). The reason for low *R*-values in aluminum and in FCC metal sheets in general is that the <111> // ND fiber texture, which is favorable for improving *R*-value, cannot be obtained via conventional rolling and annealing processes in aluminum. Poor *R*-value has remained a big challenge for aluminum sheet producers in industry. One possible solution to this problem is to develop metal forming processes that can introduce a texture different from the rolling texture. Simple shear texture is a good candidate, with shear plane parallel to the plane of the produced sheet, because it contains the <111> // ND fiber [25], for both f.c.c. and b.c.c. crystal structures. Adopting this approach, several advanced metal forming techniques have been developed to improve *R*-value in aluminum and its alloy sheets; for example, continuous shear deformation process [5], two-pass single-roll drive unidirectional shear rolling, a variant of asymmetric rolling [25], differential-friction rolling [35], and asymmetric rolling [36]. *R*-values achieved from those studies are also presented in Table 3. One can see that they are all smaller than the values obtained in Al-1050 by FALEP. Therefore, the FALEP process used in this study can be a solution to improve the *R*-values of aluminum sheets.

In order to get deeper insight into the role of the texture in the *R*-value, simulations were carried out using the polycrystal viscoplastic model. For the polycrystal behavior, the relaxed constraints Taylor-type approach was employed. The experimental texture after FALEP (Figure 6b,d) was discretized to 5000 grain orientations with the help of the ATEX software version 2.21 (Metz, France). [37]. Ideal fiber textures as well as all ideal shear texture components were also generated by the ATEX software containing each 5000 grain orientations in a 20° Gaussian spread distribution around the ideal positions. Tension test was simulated by imposing the tensile strain rate $\dot{\varepsilon }$ in direction of axis 1, while leaving the perpendicular strain rate components free (${\dot{\varepsilon }}_{22}$, ${\dot{\varepsilon }}_{33}$), that is, they were relaxed. The ${\dot{\varepsilon }}_{12}$ shear component was also relaxed, which is a shear permitted to relax by the boundary conditions for a tensile test. The *R*-value is defined by the ratio: $R={\dot{\varepsilon }}_{22}/{\dot{\varepsilon }}_{33}$. The textures were rotated by 0°, 45°, and 90° around the sheet normal axis, to obtain *R* values in these directions. The 12 {111} <110> octahedral slip systems, together with the 6 {100} <110> non-octahedral ones, were used with equal strength. For viscoplastic slip, a low value (m = 0.01) was used. The role of the {100} <110> slip systems was examined in [38]. The obtained results are presented in Table 4.

It can be seen in Table 4 that the simulated *R*-values approach well the experimental ones; nevertheless, they are smaller. The deviation can be attributed to the high sensitivity of the *R*-value to the texture components. The two fibers of shear textures, <111> // ND and <110> // SD (SD: shear direction), and all ideal shear components were examined. The simulation confirms the general view that the <111> // ND fiber texture is very advantageous for high *R*-value, with a predicted constant *R*-value of 5.7. The <110> // SD fiber has a high *R*-value only at 45°. One can see in Table 4 that for all individual ideal orientations, the predicted *R*-value is high at 45° while remaining relatively low at 90°. The *R*-value is very varying at 0° testing as a function of ideal orientation, with high contribution from the B/Bb components. The behavior of the rotated cube (R_C_) component is very anisotropic; extremely low values at 0° and 90° and a high value at 45°. As discussed in Section 4.3, this component is not an ideal component of a shear deformation texture; its origin is dynamic recrystallization. Nevertheless, as it is the second-strongest component of the texture after the C, its contribution to the *R*-value is significant; it increases the *R*-value at 45°, and decreases it at 0° and 90°. A general observation from the simulation results in Table 4 is that all shear components (plus the rotated cube) are increasing the *R*-value at 45°. This is the reason why the *R*-value is maximum at 45°. This is valid also for other tests, which are listed in Table 2—all experimental results show the maximum *R*-value at 45°.

## 5. Conclusions

The FALEP process was implemented on an ECAP machine and the Al-1050 alloy was heavily deformed for transforming its properties. The capabilities of the process for imparting large strains, significant microstructure refinement, producing simple shear texture, and remarkably improving mechanical strength and *R*-value were achieved on this alloy, at room temperature. Polycrystal simulations shed light on the reasons for obtaining high *R*-values. The following main conclusions can be drawn from the obtained results:FALEP produced sheet materials with very high and homogeneous strains, which led to significant grain refinement in Al-1050 by reducing the grain size more than 160 times; to 600 nm, in a single step;FALEP produced simple shear texture in Al-1050 with the shear plane parallel to the plane of the produced fin;FALEP significantly increased the material strength: the yield strength of Al-1050 increased by about 10 times and the ultimate strength increased about three times;FALEP produced exceptional high average *R*-value for aluminum (1.28), with moderate anisotropy ($\Delta \bar{R}$ = −0.31).Polycrystal simulations confirmed the obtained *R*-values and explained their experimental distribution, as well.

## Figures and Tables

**Figure 1 materials-14-02465-f001:**
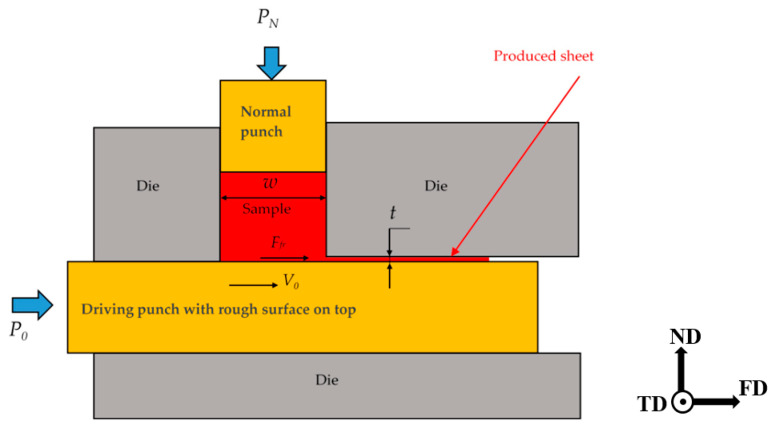
Schematic of the FALEP process.

**Figure 2 materials-14-02465-f002:**
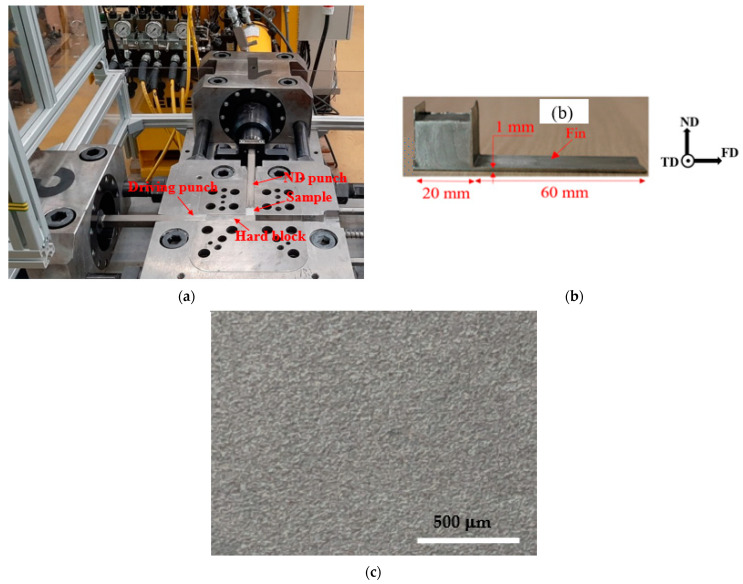
(**a**) Experimental setup, (**b**) an example of a partially extruded Al-1050 sample, and (**c**) an optical micrograph showing the quality of the bottom surface of the extruded sample.

**Figure 3 materials-14-02465-f003:**
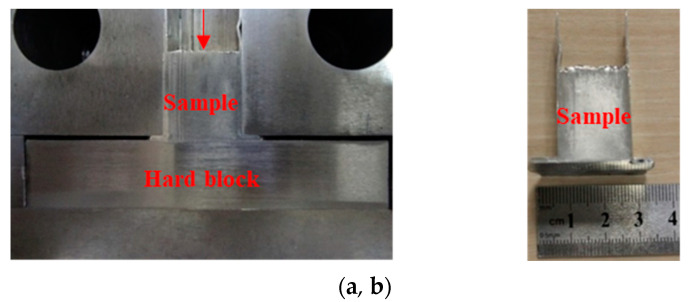
(**a**) Experimental setup for T-NECAP for Al-1050. (**b**) The sample obtained after processing.

**Figure 4 materials-14-02465-f004:**
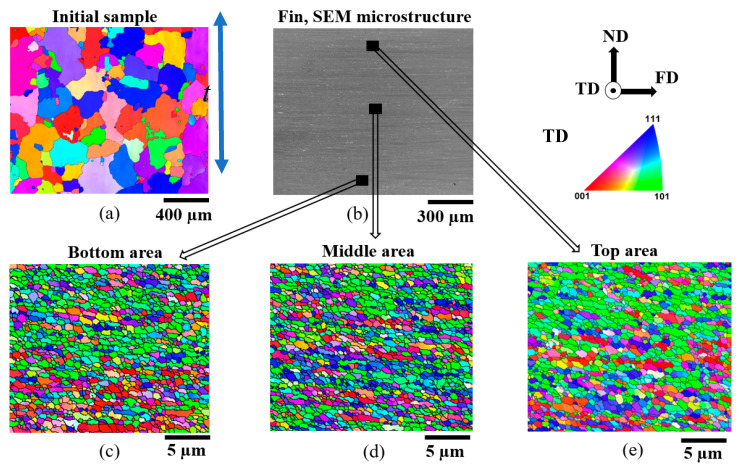
(**a**) EBSD microstructure of the initial sample; (**b**) SEM microstructure across the thickness of the produced fin; (**c**–**e**) EBSD microstructures of the bottom, middle, and top areas of the Al-1050 produced fin, respectively.

**Figure 5 materials-14-02465-f005:**
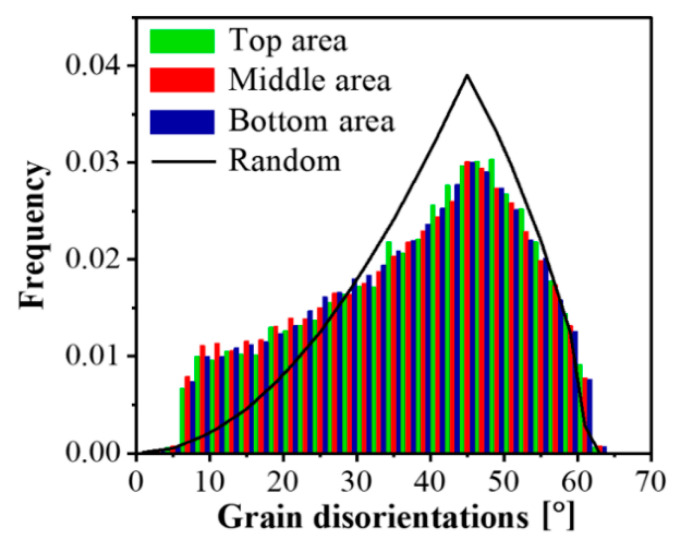
Next-neighbor grain disorientation distributions measured in the top, middle, and bottom areas of the Al-1050 produced fin.

**Figure 6 materials-14-02465-f006:**
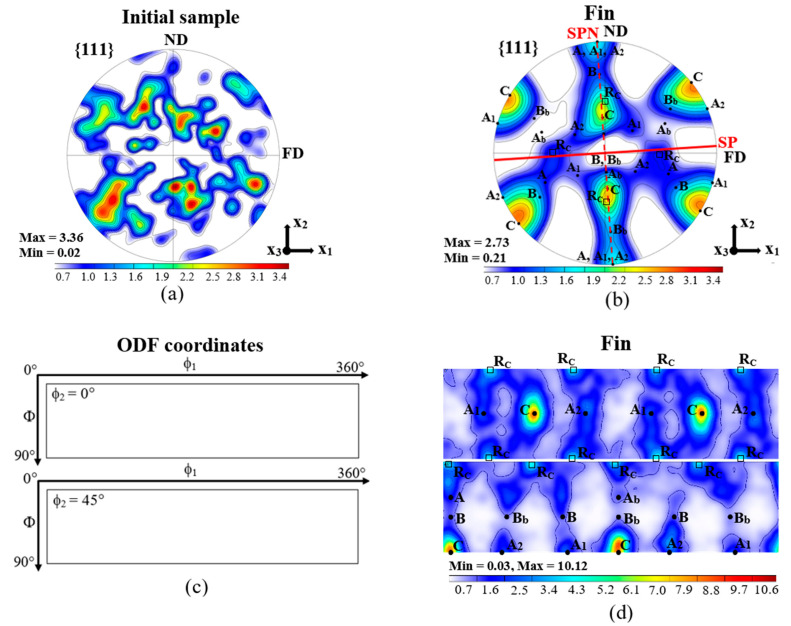
(**a**) Crystallographic texture of the initial sample, (**b**) pole figure after FALEP processing (SP: shear plane, SPN: shear plane normal), (**c**) key figure for the ODF sections, and (**d**) ODF after FALEP processing. (Rc: rotated cube component).

**Figure 7 materials-14-02465-f007:**
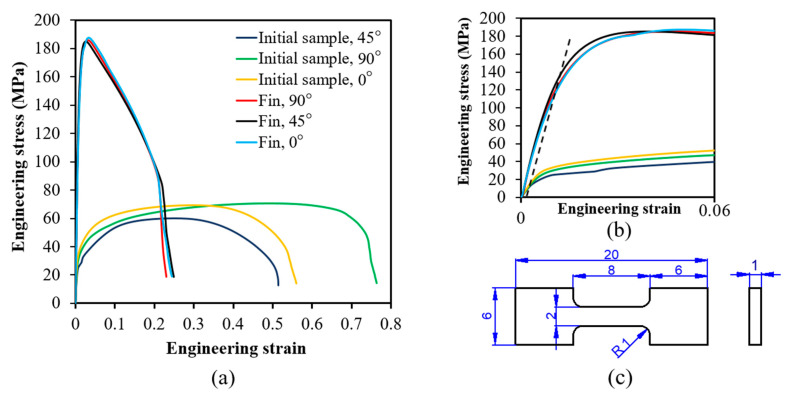
(**a**) Engineering tensile stress-strain curves of the initial sample and the FALEP-processed 1 mm thick Al-1050 fin obtained at 0°, 45°, and 90° with respect to the flow direction (FD), at room temperature. (**b**) Enlarged part of the stress-strain curve. The broken line corresponds to 0.2% plastic strain. (**c**) The geometry of the tensile specimens (mm).

**Table 1 materials-14-02465-t001:** Tensile test results on Al-1050 in initial state and after processing by FALEP. (YS: von Mises equivalent yield strength; UTS: ultimate strength).

Initial Material State					After FALEP			
Angle	YS (MPa)	UTS (MPa)	Unif. Elong. (%)	Total Elong. (%)	YS (MPa)	UTS (MPa)	Unif. Elong. (%)	Total Elong. (%)
0°	13	70	32	55	113	187	5.2	21
45°	10	60	26	52	135	185	3	22
90°	11	71	52	74	119	186	5	21

**Table 2 materials-14-02465-t002:** Measured *R*-values in aluminum.

	R_0_	R_45_	R_90_	$\bar{R}$	$\Delta \bar{R}$	Ref.
FALEP, Al-1050	1.04	1.43	1.21	1.28	−0.31	This work
Sheet ECAP, Al-1050	0.64	1.45	0.81	1.01	−0.47	[24]
Conshearing, Al-1050	0.94	-	-	-	-	[5]
Asymmetric Rolling, Al-5052	0.9	1.1	0.8	1.00	−0.14	[25]
Sheet-ECAP, AA5005	0.83	1.23	0.98	1.07	−0.65	[26]

**Table 3 materials-14-02465-t003:** Tensile test results on Al-1050 processed by different SPD processes. All tests were carried out at room temperature. (YS: von Mises equivalent yield strength; UTS: ultimate strength; PFM: Plastic Flow Machining, HPTT: High Pressure Tube Torsion, ECAP: Equal Channel Angular Pressing; ASR: Asymmetric Rolling; ARB: Accumulative Roll Bonding).

Initial Material State					After SPD Processing						
YS (MPa)	UTS (MPa)	Unif. Elong. (%)	Total Elong. (%)	Grain Size (µm)	SPD Process	YS (MPa)	UTS (MPa)	Unif. Elong. (%)	Total Elong. (%)	Grain Size (µm)	Ref.
10–13	60–71	26–52	51–76	100	1 FALEP pass	113–135	185–187	3–5.2	21–22	0.66	This paper
25–28	65–70	30–32	55–57	100	1 PFM pass	125–135	170–175	5–6	18–25	0.8–1.8	[10]
55	345	20	-	24	HPTT	346	405	2	2	0.5	[28]
-	76	35	55	-	4 Increm. ECAP	-	177	1.5	15	-	[29]
-	-	-	-	330	8 ECAP passes	-	140–175	3–3.5	10–15	0.59	[30]
90	110	-	15	-	7 ARB passes	120	150	-	9	0.7	[31]
-	100	4	17	-	60% ASR	-	176	2	6.5	-	[32]

**Table 4 materials-14-02465-t004:** Measured and simulated *R*-values.

	R_0_	R_45_	R_90_	$\bar{R}$	$\Delta \bar{R}$
Measured	1.04	1.43	1.21	1.28	−0.31
Simulated	0.95	1.26	0.98	1.11	−0.30
<111> // ND fiber	5.7	5.7	5.7	5.70	0.00
<110> // SD fiber	1.11	5.67	0.95	3.35	−4.64
A/Ab component	0.2	9.3	0.95	4.94	−8.73
B/Bb component	6.47	2.3	1.14	3.05	1.51
C component	7.9	3.6	1.4	4.13	1.05
A1/A2 component	1.83	1.29	1.48	1.47	0.37
R_C_ (45° rotated cube)	0.15	2.68	0.15	1.42	−2.53
Random texture	1.00	1.00	1.00	1.00	0.00

## Data Availability

The measurement data are available on request from the corresponding author. The software for simulating *R*-values is available with free access at: https://github.com/LaszloSToth/Polycrystal-plasticity-software.git.

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
