# Peer review of "Microstructure, Texture and Mechanical Properties in Aluminum Produced by Friction-Assisted Lateral Extrusion"

_materials, 2021, doi:10.3390/ma14092465_

Round 1

Reviewer 1 Report

Title: Microstructure, texture and mechanical properties in aluminum produced by friction assisted lateral extrusion
Authors: Viet. Q. Vu, Laszlo. S. Toth, Yan Beygelzimer and Yajun Zhao

Summary: 

In this manuscript, the authors study texture and mechanical properties of
aluminum produced by friction assisted lateral extrusion process (FALEP), a relatively new severe plastic deformation (SPD) process where the sample is extruded by combining normal force with shear force applied through friction. It is noted that the main advantage of FALEP is that it is a single step process, and results in a significant and homogeneous grain refinement (~160 times). The authors study the mechanical properties (stress vs strain) of Al-1050 processed using FALEP, and crystallographic texture and its anisotropy by measuring the Lankford parameter. The authors report a three-fold increase in strength (at the cost of ductility), and an in increase in the Lankford parameter suggesting improved formability.

Review:

The topic of the manuscript is certainly relevant to the research community and the Journal, and it is well-written with data supporting the observations. I
recommend this article for publication after the following questions have been addressed.

  1. In section 4.2, the authors observe a difference between the Mackenzie grain misorientation distribution of a random microstructure, and that of a sample processed using FALEP. They reason that this is a result of FALEP preferring certain misorientations. It is not clear which misorientations are preferred. Moreover, it is mentioned that 92% of grain boundaries (GBs) are large angle GBs. Does Figure 5 reflect this, if so how?
  2. In Section 4.5, it is stated without any reasoning that the shear texture observed in FALEP improves. Their measurements certainly reflect this, but exactly how and why shear texture effects R-value has to be explained.
  3. The caption to Fig. 6 is confusing, and it may have to be rephrased.
  4. All figures may have to be resized and centered.

Author Response

We are grateful for all four Reviewers for taking their precious time to deal with our manuscript and providing useful critical assessments. We significantly improved the paper and marked all major changes with yellow highlight. New simulation work was also done, presented in the Discussion part. We think that - thanks to the advices and our further investment - the revised version reads much better than the first one. 

Reviewer 1:

Title: Microstructure, texture and mechanical properties in aluminum produced by friction assisted lateral extrusion
Authors: Viet. Q. Vu, Laszlo. S. Toth, Yan Beygelzimer and Yajun Zhao

Summary: 

In this manuscript, the authors study texture and mechanical properties of
aluminum produced by friction assisted lateral extrusion process (FALEP), a relatively new severe plastic deformation (SPD) process where the sample is extruded by combining normal force with shear force applied through friction. It is noted that the main advantage of FALEP is that it is a single step process, and results in a significant and homogeneous grain refinement (~160 times). The authors study the mechanical properties (stress vs strain) of Al-1050 processed using FALEP, and crystallographic texture and its anisotropy by measuring the Lankford parameter. The authors report a three-fold increase in strength (at the cost of ductility), and an in increase in the Lankford parameter suggesting improved formability.

Review:

The topic of the manuscript is certainly relevant to the research community and the Journal, and it is well-written with data supporting the observations. I
recommend this article for publication after the following questions have been addressed.

  1. In section 4.2, the authors observe a difference between the Mackenzie grain misorientation distribution of a random microstructure, and that of a sample processed using FALEP. They reason that this is a result of FALEP preferring certain misorientations. It is not clear which misorientations are preferred. Moreover, it is mentioned that 92% of grain boundaries (GBs) are large angle GBs. Does Figure 5 reflect this, if so how?

Answer: We made this part clearer by rewriting the correspond part in this way:

The difference between the random (Mackenzie) distribution and the experimental one is due to the applied large deformation during which a non-random crystallographic orientation distribution develops (called texture). The fraction of large angle boundaries was very high; about 92%, which was obtained by integrating the disorientation distribution in Figure 5 for angles larger than 15°, then multiplied by 100 (the total integral is 1.0.). This fraction is the same value as for a shear strain of about 20 or more in Al-1050 deformed by other shear processing [22]. 

  1. In Section 4.5, it is stated without any reasoning that the shear texture observed in FALEP improves. Their measurements certainly reflect this, but exactly how and why shear texture effects R-value has to be explained.

Answer: The reason why shear textures have good R values is now shown by polycrystal plasticity simulations of the R values in shear. This is included into the Discussion section.

  1. The caption to Fig. 6 is confusing, and it may have to be rephrased.

Answer: Thanks for noticing this. We have reorganized the figure and the caption as well.

  1. All figures may have to be resized and centered.

Answer: We used the template now provided by the Journal, but we do not know how it is going to be printed from the template.

Reviewer 2 Report

Authors of the article designed and tested successfully the Friction-Assisted Lateral Extrusion Process (FALEP). Compared to Plastic Flow Machining, FALEP transforms not only a surface layer but the entire bulk sample into a sheet.

They examined for microstructure and the crystallographic texture using a modern JEOL JSM-6500F field-emission gun-scanning electron microscope with Electron Back Scattering Diffraction (EBSD).

The materials of the article are relevant and interesting. The text of the article would be more complete if the uniformity of plastic deformation over the thickness of the sheet was confirmed by the methods of finite element simulating.

The article can be accepted for publication in the journal Materials.

Author Response

We are grateful for all four Reviewers for taking their precious time to deal with our manuscript and providing useful critical assessments. We significantly improved the paper and marked all major changes with yellow highlight. New simulation work was also done, presented in the Discussion part. We think that - thanks to the advices and our further investment - the revised version reads much better than the first one. 

Reviewer 2:

Authors of the article designed and tested successfully the Friction-Assisted Lateral Extrusion Process (FALEP). Compared to Plastic Flow Machining, FALEP transforms not only a surface layer but the entire bulk sample into a sheet.

They examined for microstructure and the crystallographic texture using a modern JEOL JSM-6500F field-emission gun-scanning electron microscope with Electron Back Scattering Diffraction (EBSD).

The materials of the article are relevant and interesting. The text of the article would be more complete if the uniformity of plastic deformation over the thickness of the sheet was confirmed by the methods of finite element simulating.

Answer: Thank you for appreciating our work. While it is useful to carry out finite element simulations, it was not planned in this article due to the absence of sufficient expertise on FE simulations in our research group. This can be done by other researchers in future work. Nevertheless, our conclusion on the uniformity of deformation is based on the extreme uniformity of the microstructure, the grains shapes and forms, the uniformity of texture, and the extreme uniformity of the disorientation distribution. Namely, if all these experimental features are all uniform, it means the deformation is also uniform, as all these features depend on the amount of deformation.

In order to be clearer on this point, we presented the above reasoning in the discussion part.    

The article can be accepted for publication in the journal Materials.

Reviewer 3 Report

The Authors present a method of friction assisted lateral extrusion to obtain micro-structure Al sheets with improved mechanical properties.

The preparation and, in particular, the investigation tools are well presented and therefore the results are of interest in the field.

I do have one main objection however.

As a reader, I would like to find in the paper a detailed comparison between the properties of the samples obtained through this method and those obtained previously by comparable methods (described in articles of the same Authors or of other groups). Example: How do the grain size distribution/orientation or the R-value compare to the case of previous samples similarly obtained, described in existing literature. All properties should be compared.

If this objection is answered, I do not see reasons against publication of the manuscript.

Author Response

We are grateful for all four Reviewers for taking their precious time to deal with our manuscript and providing useful critical assessments. We significantly improved the paper and marked all major changes with yellow highlight. New simulation work was also done, presented in the Discussion part. We think that - thanks to the advices and our further investment - the revised version reads much better than the first one. 

Reviewer 3:

The Authors present a method of friction assisted lateral extrusion to obtain micro-structure Al sheets with improved mechanical properties.

The preparation and, in particular, the investigation tools are well presented and therefore the results are of interest in the field.

I do have one main objection however.

As a reader, I would like to find in the paper a detailed comparison between the properties of the samples obtained through this method and those obtained previously by comparable methods (described in articles of the same Authors or of other groups). Example: How do the grain size distribution/orientation or the R-value compare to the case of previous samples similarly obtained, described in existing literature. All properties should be compared.

Answer: We have added now detailed comparisons with existing findings in the field of SPD research.

If this objection is answered, I do not see reasons against publication of the manuscript.

Reviewer 4 Report

the current study evaluate the mechanical properties and microstructure of aluminium manufactured using friction assisted lateral extrusion. The authors found that the microstructure of the A1050 alloy is refined and grain size was reduced considerably. The authors also found that the ultimate yield strength increased.

there is a difference in font size in the abstract please check this issue elsewhere in the manuscript!

Why this “Lankford parameter” is smaller in size that the rest of text?

The abstract needs to be completely rewritten, please clearly state what you have done and what things you measured in terms of mechanical testing. Please consider reviewing the abstract and highlight the novelty, major findings and conclusions.

Literature review is limited and needs to be further explored and elaborated, please discuss about past studies similar to your work on extrusion of aluminium alloys or other metals, discuss what they have done and what were their main findings then highlight how does your current work brings new knowledge and difference to the field and industry.

please add mechanical properties of the studied alloy in the materials and methods section and references it if needed

why you speficially choose this alloy? 

how many times were the tests repeated? 

Please also answer the following question in the introduction, what is the research gap did you find from the previous researchers in your field? Mention it properly. It will improve the strength of the article.

Section 2 I don’t think it adds any value to your work, we already know what is FALEP process and how it work, this can be more suited for a book chapter or a student thesis but not in a scientific paper.

Please add a list/table of nomenclature at the start or end of the manuscript to summarise the meaning of all the symbols and Greek letters used in this work

Please change section 4 name to materials and methods

The authors should adhere to the guidelines of the journal template which appears is not being followed here.

Some of the data in section 4 can be better represented using a table

Combine figures 2 and 3 in one figure if possible, add some arrows and text for the images in figure 3 to tell the readers what are they looking at in these pictures.

The authors should clearly add a section called “results and discussion”

The paper layout is very poor and needs to be significantly improved

“the microstructure of the produced fin exhibited significant grain refinement with a grain size reduced more than 160 times; to 600 nm” please discuss this claim in more details and try to relate to past studies and what did they find? Was it similar to  your work or different, in any case support with references and provided detailed analysis.

The results are merely described and is limited to comparing the experimental observation. The authors are encouraged to include a discussion section and critically discuss the observations from this investigation with existing literature.

Author Response

We are grateful for all four Reviewers for taking their precious time to deal with our manuscript and providing useful critical assessments. We significantly improved the paper and marked all major changes with yellow highlight. New simulation work was also done, presented in the Discussion part. We think that - thanks to the advices and our further investment - the revised version reads much better than the first one. 

Reviewer 4:

the current study evaluate the mechanical properties and microstructure of aluminium manufactured using friction assisted lateral extrusion. The authors found that the microstructure of the A1050 alloy is refined and grain size was reduced considerably. The authors also found that the ultimate yield strength increased.

there is a difference in font size in the abstract please check this issue elsewhere in the manuscript!

Why this “Lankford parameter” is smaller in size that the rest of text?

Answer: These differences were introduced by the Template, used by the Journal. We did not use the Template, the Journal used it. We hope that it will be settled properly at the end.

The abstract needs to be completely rewritten, please clearly state what you have done and what things you measured in terms of mechanical testing. Please consider reviewing the abstract and highlight the novelty, major findings and conclusions.

Answer: The major part of the Abstract has been reformulated.

Literature review is limited and needs to be further explored and elaborated, please discuss about past studies similar to your work on extrusion of aluminium alloys or other metals, discuss what they have done and what were their main findings then highlight how does your current work brings new knowledge and difference to the field and industry.

Answer: We tried to point out these elements, in the Introduction, as well as in numerous comparisons with existing results on different extrusion processes, mostly in form of Tables.

please add mechanical properties of the studied alloy in the materials and methods section and references it if needed

Answer: It is done in the revised version.

why you speficially choose this alloy?

Answer: Al1050 is a material frequently used in applications (for example, in electricity) because of its good electrical resistivity and better strength compared to pure Al, due to the 0.5% impurity content. Further strengthening, however, is not possible by heat treatments, contrary to other Al alloy systems. One effective way is to decrease the grain size and get strengthening due to the Hall-Petch effect. Severe plastic deformation is an effective way to reduce the grain size, which can be readily realized in industrial applications if the process is simple enough and final result can be achieved in a single operation step. 

We have included this addition in the revised version.

how many times were the tests repeated? 

Answer: Many times, at least 10 times, with good reproduction of the results. It is now mentioned.

Please also answer the following question in the introduction, what is the research gap did you find from the previous researchers in your field? Mention it properly. It will improve the strength of the article.

Answer: The well-known SPD processes require either multi-passes (sheet ECAP [16], high strain rolling, ARB [3]), or produce small samples (high pressure torsion HPT [2]) for obtaining a sheet with UFG structure. This are retracting points for SPD processes. Another point is that sheets normally contain rolling textures, other kinds of textures in sheet are not frequent, especially not a perfect simple shear texture, which can be readily obtained from a bulk metal by FALEP. Therefore, the main research gap in existing SPD research is to obtain simple shear textures in sheet form from bulk material. Concerning aluminum, it is also well known that an Al sheet has low R-value, which is reducing its deep drawing properties, so one has to try to introduce other kind of textures into Al sheets to improve the R-values. So these objectives were set up for the present research.

Section 2 I don’t think it adds any value to your work, we already know what is FALEP process and how it work, this can be more suited for a book chapter or a student thesis but not in a scientific paper.

Answer: As far as we know, FALEP is a totally unknown process, although was published in 1992. However, the SPD community does not know it at all. We presented it in a recent conference but there is no detailed paper on FALEP processing since 29 years. If the Referee knows about a textbook which is presenting it, we would appreciate to have the information. Moreover, it is always necessary to describe the experiment which is used for material processing. We are convinced that without this, the paper could not stand at all. We integrated now Section 2 into the Materials and Methods section.

Please add a list/table of nomenclature at the start or end of the manuscript to summarise the meaning of all the symbols and Greek letters used in this work

Answer: We are using very limited notations, there are only two simple equations in the paper, so we think that such kind of list is not justified for this paper; it is usual only in cases where there are many equations and variables. The meanings of the notations are clearly specified in the text.

Please change section 4 name to materials and methods

Answer: We have reorganized the Sections. Now there is a Materials and Methods section.

The authors should adhere to the guidelines of the journal template which appears is not being followed here.

Answer: Yes, we did not use the template (it is not obligatory): it is the Journal who used it. We also find that the produced layout looks quite bad.

Some of the data in section 4 can be better represented using a table

Answer: Yes, we made several new tables.

Combine figures 2 and 3 in one figure if possible, add some arrows and text for the images in figure 3 to tell the readers what are they looking at in these pictures.

Answer: Figure 3 is presenting another test, so it is desirable to have it separate. We added some text in Figure 3 for better presentation.

The authors should clearly add a section called “results and discussion”

Answer: Agree. Now the sections are called: 2. Materials and Methods, 3. Experimental results, 4. Discussion.

The paper layout is very poor and needs to be significantly improved

Answer: Agree, as discussed above. We are also asking the Journal to provide better template.

“the microstructure of the produced fin exhibited significant grain refinement with a grain size reduced more than 160 times; to 600 nm” please discuss this claim in more details and try to relate to past studies and what did they find? Was it similar to  your work or different, in any case support with references and provided detailed analysis.

The results are merely described and is limited to comparing the experimental observation. The authors are encouraged to include a discussion section and critically discuss the observations from this investigation with existing literature.

Answer: Now there is a detailed Discussion section with critical interpretation of the obtained results and comparison with other publications. New polycrystal plasticity simulation work was also carried out which provided an interpretation of the R-value results.

Round 2

Reviewer 4 Report

All questions answered paper can be accepted